# Transcriptional Response of Circadian Clock Genes to an ‘Artificial Light at Night’ Pulse in the Cricket *Gryllus bimaculatus*

**DOI:** 10.3390/ijms231911358

**Published:** 2022-09-26

**Authors:** Keren Levy, Bettina Fishman, Anat Barnea, Amir Ayali, Eran Tauber

**Affiliations:** 1School of Zoology, Tel Aviv University, Tel-Aviv 6997801, Israel; 2Department of Evolutionary and Environmental Biology, Institute of Evolution, University of Haifa, Haifa 3498838, Israel; 3Department of Natural and Life Sciences, The Open University of Israel, Raanana 4353701, Israel; 4Sagol School of Neuroscience, Tel Aviv University, Tel-Aviv 6997801, Israel

**Keywords:** light pollution, artificial light at night, ALAN, insects, circadian rhythm, extracellular RNA

## Abstract

Light is the major signal entraining the circadian clock that regulates physiological and behavioral rhythms in most organisms, including insects. Artificial light at night (ALAN) disrupts the natural light–dark cycle and negatively impacts animals at various levels. We simulated ALAN using dim light stimuli and tested their impact on gene expression in the cricket *Gryllus bimaculatus*, a model of insect physiology and chronobiology. At night, adult light–dark-regime-raised crickets were exposed for 30 min to a light pulse of 2–40 lx. The relative expression of five circadian-clock-associated genes was compared using qPCR. A dim ALAN pulse elicited tissue-dependent differential expression in some of these genes. The strongest effect was observed in the brain and in the optic lobe, the cricket’s circadian pacemaker. The expression of *opsin-Long Wave* (*opLW*) was upregulated, as well as *cryptochrome1-2* (*cry*) and *period* (*per*). Our findings demonstrate that even a dim ALAN exposure may affect insects at the molecular level, underscoring the impact of ALAN on the circadian clock system.

## 1. Introduction

For most animal species, light detection is vital for their temporal adaptation to the earth’s diel cycles. Monitoring the annual change in the daily light duration (photoperiod) is also used by many organisms in adapting to the seasonal cycle and conditions [1,2,3]. Many behaviors depend on such cycles: daily periods of activity and rest, sleep, foraging, courtship and mating, ecdysis in insects, migration, as well as diapause, to name just a few [1,4,5,6,7]. Consequently, many living organisms have evolved sensitive mechanism for light detection, and utilize light for other functions in addition to vision, such as entrainment of the circadian clock, timing of gene expression, sexual maturation, and hormonal regulation [4,5,8,9,10].

Artificial light at night (ALAN) is a fast-growing worldwide phenomenon [11,12]. ALAN refers to both an increase in light intensity and changes in the naturally occurring light spectrum. ALAN causes changes in the behavior and temporal activity of many animals, such as birds [13,14], rodents [15], anurans [16], and insects [17,18]. ALAN impairs sleeping behavior [19], camouflage, and population synchronization [17,20,21]. Moreover, it affects predation [22,23], orientation [24], and pollination [25,26], causes high insect mortality [27,28,29], changes in community structures, and changes in biodiversity [30,31,32]. Although the specific impacts of ALAN on insects have recently received considerable attention [33], the underlying molecular mechanisms are still poorly understood. Here, we specifically focus on the effect of dim ALAN on the circadian system, which has been well studied in vertebrates [34,35,36], but to a lesser extent in insects.

The circadian clock and its light input pathways in insects have been studied extensively in the fruit fly, *Drosophila melanogaster* [37,38], as well as in cockroaches, crickets, and bees (reviewed in [39]). The cricket *Gryllus bimaculatus* has been widely used as a model for behavior and circadian activity [9,40]. In contrast to the fruit fly, whose clock neurons are located in the brain, the circadian pacemaker of the cricket is located in the optic lobes [39,41,42,43,44,45,46,47,48]. Light entrainment of the circadian system in the cricket is mediated by green-sensitive opsins in the insect’s compound eyes [43]. The pacemaker consists of two major feedback loops, one of which is based on the *period* (*per*) and *timeless* (*tim*) genes, repressing their own transcription by inhibiting the transcription of *Clock* (*Clk*) and *cycle* (*cyc*). The other feedback loop is based on the two *cry* genes, *cry1* (a *Drosophila*-type *cry*), which is a light sensitive photopigment, and *cry2*, a light insensitive mammalian-type *cry* [39,49]. This second feedback loop involves the upregulation of *PAR domain protein 1* (*Pdp1*) and *c-fosB* [49,50,51].

Recently, we have demonstrated that the lifelong exposure of male crickets to dim ALAN as low as 2 lx leads to a loss of rhythmicity and desynchronization of stridulation and locomotion behavior [17]. Stridulation serves to attract females and is therefore crucial for the species’ fitness. We have reasoned that these behavioral changes are mediated by changes in gene expression. Indeed, recent studies on birds [52,53], amphibians [54], and glow-worms [55] have revealed altered gene expression following exposure to dim ALAN, an effect that was present in both the visual system and various other tissues. Here, we sought to investigate the effect of exposure to dim ALAN on gene expression in four different tissues of the cricket *G. bimaculatus*. Due the effect of ALAN on male behavior (locomotion, stridulation), we have focused on transcriptional response in males. Understanding the effect of dim ALAN on internal processes and pathways is crucial for assessing the threats of ALAN to pollinators and ground-dwelling insects, as well as for predicting the long-term effects on ecosystems.

## 2. Results

The relative levels of gene expression in the different cricket tissues and hemolymph samples following a 30 min light pulse of 2, 5, or 40 lx, and a no-pulse control, are shown in Figure 1. In the brain, the expression of most genes was affected by the different light intensities (Figure 1A). The expression levels of both *per* and *cry2* significantly increased with increasing light pulse intensity (F_3,21_ = 3.70, *p* = 0.028, and F_3,24_ = 3.04, *p* = 0.048, respectively, Figure 1A). A dependence on light pulse intensity was also observed in *cry1*, although the effect was only marginally significant (F_3,21_ = 2.96, *p* = 0.056). The expression of *opLW* differed significantly only when light treatment experiments were pooled and compared with the control (F_1,26_ = 4.90, *p* = 0.036). The *c-fosB* transcription in the brain did not differ among the different treatments (F_3,24_ = 1.15, *p* = 0.35).

In the optic lobe, the expression of *opLW* was 100-fold higher compared to that seen in the brain but did not change significantly following exposure to light (Figure 1B). In contrast to the brain (Figure 1A), in which most genes exhibited light-induced upregulation, in the optic lobe the light stimuli elicited a decrease in the expression of most genes. Downregulation was observed in *cry2* although only marginally significant (F_3,16_ = 2.86, *p* = 0.07) and a similar trend was observed in *cry1* (Figure 1B). No change in *per* and *c-fosB* expression was observed (F_3,17_ = 1.45, *p* = 0.26, F_3,18_ = 0.98, *p* = 0.42, respectively).

In the Malpighian tubules most gene expression did not change with increased light pulse intensity (Figure 1C). A marginally significant effect was observed for *opLW* (F_3,42_ = 2.58, *p* = 0.066) and no differences were found for *cry2* (F_3,42_ = 0.80, *p* = 0.501), *per* (F_3,40_ = 1.14, *p* = 0.34), or *c-fosB* (F_3,42_ = 0.18, *p* = 0.91). However, *opLW* expression was higher in the control, compared to the light treatments pooled together (F_1,44_ = 6.84, *p* = 0.012), and the expression of *cry1* significantly diminished following exposure to any of the light pulses (F_3,41_ = 7.21, *p* = 0.0005, Figure 1C).

Interestingly, we found traces of expression of all genes in the hemolymph (Figure 1D). Gene expression did not differ significantly among samples. However, when light treatments were pooled together, upregulation in *per* was observed (F_1,18_ = 5.83, *p* = 0.027, Figure 1D).

In order to enable a better comparison of gene expression among tissues, we repeated the analysis using within-tissue normalization of the expression values (relative to the control mean expression values in each tissue, Figure 2). Gene expression patterns varied substantially among tissues. The expression of *opLW* was strongly upregulated in the brain and in the optic lobe throughout all three light treatments, while being downregulated in the Malpighian tubules (Figure 2A). The light stimuli elicited upregulation of *cry2* in the brain and downregulation in the optic lobe (Figure 2B). The transcriptional response of *per* showed a similar pattern, being upregulated in the brain and downregulated in the Malpighian tubules (Figure 2C). The expression of *c-fosB* showed no significant changes in the brain, while being downregulated in the optic lobe. *c-fosB* was the only gene that was upregulated in the Malpighian tubules (Figure 2D). Expression of *cry1* was downregulated in all tissues except the brain, where it was upregulated under 40 lx only (F_3,23_ = 7.77, *p* < 0.001. Figure 2E). The PCA of the relative gene expression clearly separated the optic lobe and the hemolymph from the brain and the Malpighian tubules, while the latter two mostly overlapped (Figure 2F).

To simultaneously analyze the transcriptional differences between the treatments, we used Linear Discriminant Analysis (LDA). The first two LD functions explained 67.12% of the variation in the brain, 73.29% in the optic lobes, 66.87% in the Malpighian tubules and, 73.76% in the hemolymph (Figure 3). In the brain, the control group was clearly separated from the 5 and 40 lx treatments, but not from the 2 lx one (Figure 3A). In contrast, the clustering of the treatments in the optic lobe revealed a substantial difference between the control and all ALAN treatments (Figure 3B). In the Malpighian tubules and in the hemolymph, the discriminant map did not reveal any significant difference between the control and the ALAN treatments (Figure 3C,D).

## 3. Discussion

Insects have served for almost a century as successful models in laboratory studies on the role of light in regulating circadian behavior [7], and in raising questions about the underlying molecular mechanisms. Here, we investigated the effect of a short, dim ALAN pulse on the gene expression of five genes associated with the circadian system of the cricket *Gryllus bimaculatus*.

This work substantiates the findings of previous studies [56] that uncovered the heterogeneity of circadian pacemakers across various tissues in *G. bimaculatus* (see PCA analysis, Figure 2F), and demonstrates the tissue-specific effects of ALAN on transcription. In the optic lobe, where the cricket’s central pacemaker is located [46,48,57,58], the response to ALAN exposures was clearly distinct from that of the control, presumably due to the strong effect of dim ALAN on *opLW* expression. In the brain, transcriptional responses of the control and the 2 lx treatment were largely overlapping, and to a lesser extent also with those of the 5 lx and 40 lx treatments. The Malpighian tubules and hemolymph seemed less affected by the light exposures. Indeed, a previous study [56] suggested that *G. bimaculatus* tubules do not harbor a circadian pacemaker, as *per* transcript levels lacked any diurnal rhythm (in contrast to in *Drosophila* [59]).

The expression of circadian clock genes in *G. bimaculatus* in tissues other than the optic lobes has been previously reported [56]. Particularly interesting is the role of these genes in the brain. Expression of *per* oscillates under constant darkness (DD), and this oscillation persists after removal of the optic lobes. However, in the absence of the optic lobes, the phase of the brain *per* mRNA rhythm is aberrant. These results allude to a brain circadian pacemaker that is subordinate to the central clock in the optic lobes [56].

In the hemolymph, we found minute levels of circadian clock transcripts, which to the best of our knowledge is the first report of extracellular RNA (exRNA) in circadian clock transcripts. exRNA is emerging as a newly discovered form of intracellular signaling [60]. It is secreted to the biofluids, encapsulated by protecting extracellular vesicles. The exRNA is largely composed of small non-coding RNA, and to a lesser extent messenger RNA [60]. exRNA have been shown to be an effective biomarker for diagnostic purposes, but their functional role awaits further study.

The rather rapid transcriptional response to light of clock genes such as *per* or *cry*, is intriguing, and the underlying molecular mechanism warrants further investigation. In *Drosophila*, a similar rapid change of *per* mRNA following a light pulse at *zeitgeber* time (ZT) 15 has been previously reported [61,62], although, in contrast to the cricket, light evoked downregulation of *per.* It was suggested that the response to light is mediated by an histone acetylation mechanism. In accord with studies in other insects (e.g., ants [63] and lepidoptera [64]), we observed very high levels of *opLW* in the crickets’ optic lobe. All ALAN treatments induced some upregulation in *opLW* expression in the optic lobe, although not significant (we sampled 4.5 h post lights-off, while 6 h after onset of darkness a substantial downregulation has been reported [43], Figure 1 therein). Given that the circadian photosensitivity of *G*. *bimaculatus* varies during the day, as was shown by the phase response curve of this species [65,66], one can predict that the effect of transient ALAN would be time dependent.

The disruption caused by ALAN to the circadian transcriptional cycle (Figure 1) is likely to disrupt the diurnal pattern of behavior (locomotor, stridulation), but the specific behavioral output awaits further study.

The circadian disruption caused by dim (2–5 lx) ALAN may have ecological implications, which should be taken into consideration when designing regulations concerning suitable light intensity for park and city lights [67]. Not only flying insects, which are attracted to light [28], are threatened by ALAN, but also ground-dwelling insects [21,24,68]. The deep-level effect of ALAN may therefore threaten many more species of insects than currently assumed, and add to the ongoing insect decline described in the last century [69,70]. Our own work in crickets [17] demonstrated that ALAN can lead to arrhythmicity in locomotory behavior and male stridulation and therefore may affect the survival and reproduction of exposed individuals.

It should also be noted that outdoors, especially in proximity to cities and streetlights, insects are not exposed to a single light pulse, but rather to night-long, and even lifelong ALAN [17]. Populations of the small ermine moth for example, were found to reduce their flight-to-light behavior following long-term exposure to ALAN, which may result in lower predation risk and mortality in this population [71]. It remains unknown whether crickets, and indeed other insects, have developed similar adaptations to chronic ALAN exposure.

## 4. Materials and Methods

### 4.1. Insect Rearing Conditions

*Gryllus bimaculatus* crickets were reared under a constant temperature of 26 ± 1 °C and white fluorescent light (5W white CFL bulb, NeptOn, 6500 K, 380–780 nm, peak: 547 and 612 nm, Appendix A), under a 12 h light:12 h dark cycle. Actual light intensity measured at the top of the containers ranged from 250 to 350 lx, while the intensity under the provided shelter (egg cartons) ranged from 20 to 60 lx. Crickets were fed three times a week with dog-food pellets and vegetables. The rearing boxes contained water flasks with absorbent cotton wool.

### 4.2. Light-Pulse Experiments and Sample Preparation

Individual males, 3–5 days post adult emergence, were removed from the breeding colony and maintained individually, under similar 12 h light:12 h dark conditions as above, for three days. On the fourth night, three hours post lights-off (i.e., at *zeitgeber* time ZT 15), the experimental animals were exposed to a 30 min long light pulse of 2, 5, or 40 lx (by partially covering the same 5 W white CFL bulb as above). An additional group (no pulse) served as control.

One hour after pulse termination (ZT 16.5), the insects were decapitated under red light (led, 600–670 nm, peak: 642 nm), and their brain, optic lobes, and Malpighian tubules were removed over ice. Hemolymph (25 µL) was also extracted from each animal. Samples were placed separately into individually marked PCR tubes containing 100 µL RNAlater (Thermo Fisher Scientific), and immediately frozen at −20 °C.

Total RNA was extracted from each sample using the Purelink RNA mini kit (Invitrogen), according to the manufacturer’s instructions. Samples with genomic DNA residuals were treated with the PureLink™ DNase Set (Invitrogen, Waltham, MA, USA). RNA concentration was determined by NanoDrop One (Thermo Fisher Scientific, Waltham, MA, USA). First strand cDNA synthesis was carried out using the RT Random Primers of the High-Capacity cDNA Reverse Transcription kit (Applied Biosystems™, Waltham, MA, USA) and treated with RNase Inhibitor (Applied Biosystems™) according to the manufacturer’s instructions. cDNA was synthesized out of 20 ng/µL RNA and was then 10× diluted.

The number of successfully collected samples per experimental group varied as follows: 5–11 brain samples, 4–6 optic lobe samples, 9–15 Malpighian tubules, and 5–7 hemolymph samples. A detailed list of the final sample sizes is given in Appendix A.

### 4.3. Primers and qPCR

The expression of two housekeeping genes (*Actin* and *Rpl18a*) and five target genes (*opLW*, *per*, *cry1*, *cry2*, *c-fosB*) was determined in all samples. Primer sequences are listed in Appendix A. A standard curve was generated for each of the genes by using a serial dilution (1:2) of the pooled cDNA [72]. qPCR (Quantstudio 3, Thermo Fisher Scientific) was run using 59 °C as annealing temperature. Fast SYBR^®^ Green Master Mix (Applied Biosystems™) was used. The cycle point was calculated using QuantStudio™ Design and Analysis Desktop Software version 1.5.1 (Applied Biosystems).

### 4.4. Data-Processing and Statistical Analysis

Gene expression was quantified using the relative standard curve method and following primer efficiencies: *opLW*: 94.38%, *cry2*: 105.95%, *per*: 100.16%, *c-fosB*: 105.02%, and *cry1*: 93.57%. The target gene expression was normalized by dividing its values by the corresponding geometric mean of both housekeeping genes. The expression values were log transformed and a nested ANOVA was applied using R version 4.1.3 [73] and the *nlme* package [74]. A mixed model was used, with light treatment and cDNA replicate as fixed and random effects, respectively. A principal component analysis (PCA) was applied to all expression values and the first two principal components (PC1 and PC2) of the PCA were used as explanatory variables for a linear discriminant analysis (LDA), with the treatments as response variables.

## Figures and Tables

**Figure 1 ijms-23-11358-f001:**
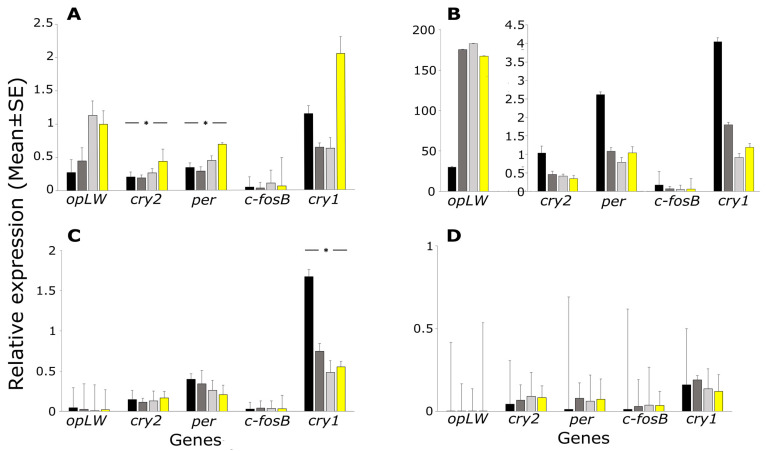
Transcriptional response of *G. bimaculatus* to dim ALAN. The relative gene expression (mean ± s.e.) following 30 min light pulse of 2 lx (dark grey), 5 lx (light grey) or 40 lx (yellow), and no-pulse control (black). Results are shown for the brain (**A**), optic lobe (**B**), Malpighian tubules (**C**), and hemolymph (**D**). * *p* < 0.05.

**Figure 2 ijms-23-11358-f002:**
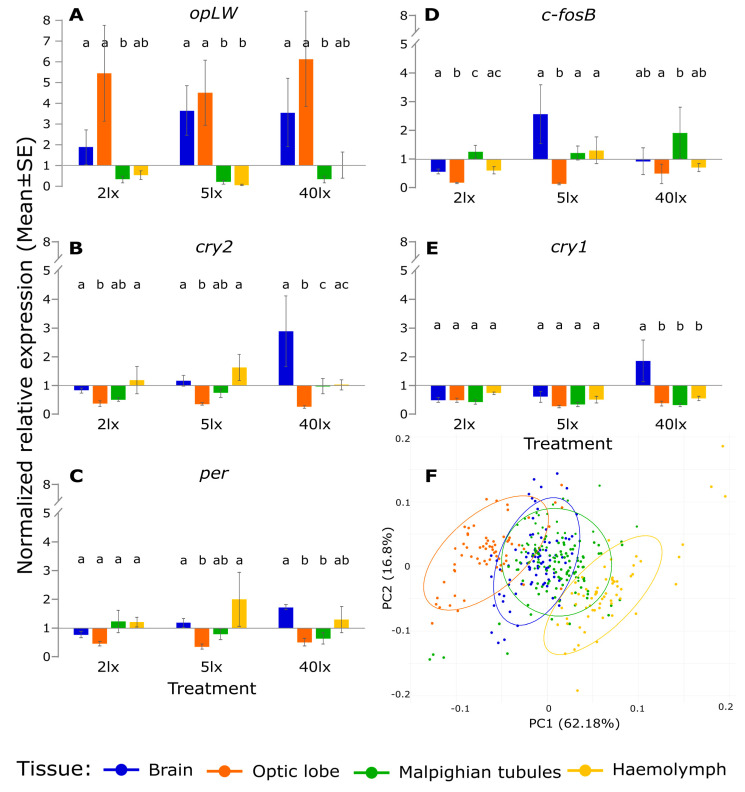
Normalized mean (±s.e.) of the relative gene expression of (**A**) *opLW*, (**B**) *cry2*, (**C**) *per*, (**D**) *c-fosB*, (**E**) *cry1,* and (**F**) Principal component analysis (PCA) of relative gene expression in the brain (blue), optic lobe (orange), Malpighian tubules (green), and hemolymph (yellow). Within-tissue normalization was performed relative to the mean of control. Different letters indicate significant differences. In the PCA, each point represents one individual sample and ellipses contain 95% of the group. PCA plot is illustrated on the two principal axes, explaining 78.98% of the variability.

**Figure 3 ijms-23-11358-f003:**
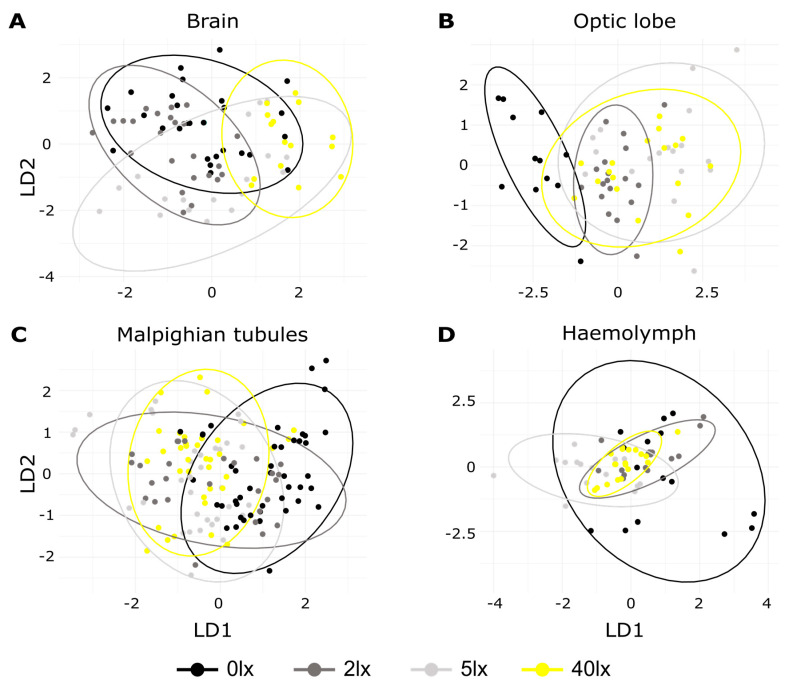
Linear discriminant analysis of transcriptional response to ALAN. The analysis was carried out under four different light treatments: 0 lx (black), 2 lx (dark grey), 5 lx (light grey), 40 lx (yellow); sampled in the brain (**A**), optic lobe (**B**), Malpighian tubules (**C**), and hemolymph (**D**). Each point represents one individual sample while ellipses represent a 95% confidence level. Each plot is illustrated on the two principal axes of the LDA, explaining >65% of the variability.

## Data Availability

The data presented in this study are openly available in FigShare at doi: 10.6084/m9.figshare.20721265 [75].

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
