# Peer review of "Transcriptional Response of Circadian Clock Genes to an ‘Artificial Light at Night’ Pulse in the Cricket Gryllus bimaculatus"

_ijms, 2022, doi:10.3390/ijms231911358_

Round 1

Reviewer 1 Report

In the paper “Transcriptional response of circadian clock genes to an 'artificial light at night' pulse in the cricket Gryllus bimaculatus” the aim of the experiments is clearly stated already in the title and the obtained molecular data are well presented and described. There are just a few minor aspects/typos that I would like the author to clarify or correct if needed.

Line 92: “observed in cry1 (n=2, Figure 1 B)”; I can’t understand the meaning of “n=2”

Figure 1 D: differences in the per expression are described as significant in the main text but there are no brackets to highlight this as in other parts of the same graph; is that an oversight?

At the end of the Results section there are a few considerations on the data presented in Figure 3 but none regarding the haemolymph. I guess they are the same as for Malpighian tubules, but I would say this explicitly. Also, in Figure 3’s caption the treatment colours don’t correspond to those in the figure (at least on my screen) and this made the figure a bit difficult to understand.

Line 153-154: “the analysis was carried out under”

Line 157: “principal axes of the PCA”; isn’t that LDA instead of PCA?

Line 161-162: “studies on the role of light in regulating”

Line 177-178: “responses of the control and the 2 lx treatment”

Line 216: number 6 is in bold but should not be

For sampling, only male crickets are picked but to me there is no obvious reason to exclude females. There is a reference to stridulating behaviour which belongs to males only and that is affected by ALAN if looking at it. I guess the ecological importance of stridulation in mating is the reason. If that is the case, please write it explicitly or provide any reason for the sample choice.

Reviewer 2 Report

This study addresses the transcriptional consequences of exposure to artificial light at night in the circadian system in an insect, the cricket Gryllus bimaculatus. This species is an important insect model for neurophysiology of circadian mechanisms and has been studied for neuronal, molecular, and behavioural aspects of the biological clock. Besides the crucial effect of clock entrainment, light perception can have highly relevant, disturbing effects on the biology of organisms (light pollution).

Focussing on the molecular physiology of transcriptional changes, five genes in the circadian control system of the cricket are analysed in this study. The transcription level was compared in five tissues, including the brain, the optic lobe housing the circadian pacemaker, or the haemolymph. The experiments are carefully planned with increasing light intensities, and use runs with control groups not exposed to night light pulses as well as runs analysed for normalised effects between the different tissues. The manuscript is well written and concise in the presentation. It expands the knowledge on the physiological effects of artificial light in an important model species and reports the first findings on exRNA of clock transcripts in hemolymph. This is an interesting work on an important topic in neuroscience and insect physiology, and I have only few comments to the authors to explain the findings in the broader context:

For the cricket, the behavioural role of artificial light at night are mentioned only in rather general terms. The introduction also addresses the topic broadly for different animals (l. 43), could you specify here the behaviours in the cricket under circadian control and provide a stronger link of your findings to behaviour in the discussion?

As the neuronal clock system is well documented to locate in the optic lobes (l. 56, 175), could you elaborate more on the distinct findings of circadian clock transcripts in the brain – what role would they have, are they subordinate systems to the clock?

As an additional point for the discussion, would you consider the timing of the ALAN relevant – e.g., would you expect other or larger effects when the light stimuli where delivered later during the dark phase?

Fig. 2: also the information is included in the figure legend, it would be helpful to include the gene name in the different graphs besides the letters for a quick overview of the different genes, especially as this panel is more complex than Fig. 1.

Fig. 3 has a low contrast for the "5lx" regime, please consider different colour format. Also, the colour coding in the figure does not match with the description provided in the figure legend.

l. 20 explain abbreviation LD in the abstract, or avoid abbreviations here

l 54 reviewed in…?

l. 87 consider including references to the figures for the optic lobe (Fig. 1B) and brain (Fig. 1A) in the text

l. 181 delete first comma

l. 187 can you provide reference, or is this novel data?

Several references seem to lack article numbers (l. 312, 320, 329, 337, 415, 455, 465), please confirm the format

l. 316   please check range of pages (0-5 cited)

l. 392   sigillatus

l. 399   Cornell

l. 430, 432 article lacks details (volume, pages)

l. 449 please check volume of article

Reviewer 3 Report

The document is well written and contains relevant information on the effect of light changes and the expression of related genes. Results clearly indicate that the response to light depends on the expression of some genes and that this expression is differential depending on the type of organ. It highlights the effect this could have on individuals in a fully enlightened world.

The authors show that an exposition for 30 20 minutes to a light pulse of 2-40 lx, even a dim ALAN exposure may affect insects at the molecular level, underscoring the detrimental impact of ALAN on the circadian clock system. This indicates the importance of reconsidering the excess of light as a factor that modifies behavior and this is imprinted on the expression of some genes.

The document contains a few observations, which are in the pdf document.
